# The Characters of Non-Coding RNAs and Their Biological Roles in Plant Development and Abiotic Stress Response

**DOI:** 10.3390/ijms23084124

**Published:** 2022-04-08

**Authors:** Xu Ma, Fei Zhao, Bo Zhou

**Affiliations:** 1Key Laboratory of Saline-alkali Vegetation Ecology Restoration, Northeast Forestry University, Ministry of Education, Harbin 150040, China; maxu990620@nefu.edu.cn; 2College of Life Science, Northeast Forestry University, Harbin 150040, China; 3Horticulture Science and Engineering, Shandong Agricultural University, Taian 271018, China

**Keywords:** miRNA, lncRNA, development, environmental regulation, abiotic stress, regulatory network

## Abstract

Plant growth and development are greatly affected by the environment. Many genes have been identified to be involved in regulating plant development and adaption of abiotic stress. Apart from protein-coding genes, more and more evidence indicates that non-coding RNAs (ncRNAs), including small RNAs and long ncRNAs (lncRNAs), can target plant developmental and stress-responsive mRNAs, regulatory genes, DNA regulatory regions, and proteins to regulate the transcription of various genes at the transcriptional, posttranscriptional, and epigenetic level. Currently, the molecular regulatory mechanisms of sRNAs and lncRNAs controlling plant development and abiotic response are being deeply explored. In this review, we summarize the recent research progress of small RNAs and lncRNAs in plants, focusing on the signal factors, expression characters, targets functions, and interplay network of ncRNAs and their targets in plant development and abiotic stress responses. The complex molecular regulatory pathways among small RNAs, lncRNAs, and targets in plants are also discussed. Understanding molecular mechanisms and functional implications of ncRNAs in various abiotic stress responses and development will benefit us in regard to the use of ncRNAs as potential character-determining factors in molecular plant breeding.

## 1. Introduction

Plants are sessile organisms, and they have evolved sophisticated regulatory mechanisms to maintain their development and overcome environmental stress, such as high and low temperatures, UV (Ultraviolet) radiation, drought, and salinity [1,2]. The functional genes determine the plants’ growth, development, and adaption to abiotic stresses. During the process of gene expression, only approximately 2% of transcribed RNA in the eukaryotic genome encodes functional proteins [3]. With the development of functional genomics and transcriptome sequencing by RNA-seq, a large number of RNAs that do not code functional proteins, known as non-coding RNAs (ncRNAs) (rRNA, tRNA, and snRNA in spliceosome; and regulatory ncRNAs), have been identified to be involved in various developmental and stress responses [4,5,6]. The regulatory ncRNAs are classified as microRNAs (miRNAs), small interfering RNAs (siRNAs), and long ncRNAs (lncRNAs, >200 nt long) in plants [7]. MiRNAs can target lncRNAs to produce phased small interfering RNAs (phasiRNAs). Conversely, lncRNAs can also serve as the origin of miRNAs or regulate the expression of miRNAs [8].

Recently, thousands of ncRNAs have been identified in plants such as *Arabidopsis* [9,10], *Brachypodium* [11], Ginseng rusty root symptom [12], sugar beet [13,14], rice [15,16], maize [17,18], wheat [19,20], soybean [21,22], tomato [23,24], and *Brassica* [25,26]. In *Arabidopsis*, tasiRNA-ARF (Trans-acting short-interfering RNA-auxin response factor) maintains the normal morphogenesis of flowers under drought and high-salinity stress conditions [27]. In *Brassica rapa*, two lncRNAs target miR160 and function in pollen development [25]. In pigeonpea, Csa-lncRNA_1231 targets Csa-miRNA-156b and regulate the expression of *SPL-12* (*SQUAMOSA PROMOTER BINDING PROTEIN-LIKE 12*), which is involved in flower development regulation [28]. In *Prunus mume*, lncRNA XR_514690.2 downregulates ppe-miR172d, and it upregulates *AP2* (*APETALA2*), which is related to flower development [29]. In addition, the expression of lncRNA973 in cotton increases under salt-stress conditions, and in lncRNA973-overexpressed *Arabidopsis*, the tolerance of salt stress is also enhanced [30]. Moreover, miR393a, miR156d, and miR172b regulating *HvTIR1/HvAFB2* (*TRANSPORT INHIBITOR RESPONSE 1/AUXIN SIGNALING F-BOX 2*), *UGTs* (*UDP-SUGAR GLYCOSYLTRANSFERASES*), and *HvAP2* are responsible for salt tolerance in the roots of barley, and miR319a/miR396e module, miR159a and miR172b regulating *GRFs* (*GROWTH-REGULATING FACTORS*), *MYB33*, and *HvAP2* might also contribute to salt tolerance in the shoots of barley [31]. Thus, the ncRNAs, acting as important regulators, are involved in the interplay of molecular regulatory pathway of plant development and abiotic stress responses.

With the development of molecular biology and high-throughput sequencing technology, numerous functional genes involved in plant development and abiotic stress responses have been identified, and the mysterious veil of ncRNAs is also being gradually uncovered. Many ncRNAs have been the focus of several review articles published in recent years [4,32,33,34,35,36]. Understanding ncRNA-guided development and stress regulatory networks can provide new insights to improve plant tolerance to environmental stresses. However, the plant ncRNA regulatory pathways are complex interaction networks and the identical ncRNA may serve as important regulator in both plant development and various abiotic stress responses. The molecular mechanisms and the various regulatory roles of ncRNAs and their targets remain to be deciphered. Recent studies have revealed that the genetic transformation of ncRNAs and their target genes can change the phenotypes and tolerance of abiotic stresses in plants. In this review, we summarize the current advances on ncRNAs, with a focus on classification, molecular regulation mechanism, and the regulatory network in the development and abiotic stress response in plants.

## 2. Classification and Action of Plant ncRNAs

Plant ncRNAs originate in the intergenic or intronic regions of chromosomal DNA and regulate the expression of growth and development; and biotic- and abiotic-stress-response-related genes at the transcriptional, posttranscriptional, and epigenetic level in plants [5,37]. According to the molecular structure, plant ncRNAs are classified into linear ncRNAs and circular ncRNAs. Linear ncRNAs can be divided into housekeeping ncRNAs, which include ribosomal RNA (rRNA), transfer RNA (tRNA), small nuclear RNA (snRNA), and small nucleolar RNA (snoRNA) and regulatory ncRNAs, which comprise small RNAs (sRNAs) and long ncRNAs (lncRNAs) according to molecular function [37]. In plants, sRNAs comprise miRNAs, natural antisense transcript-derived small interfering RNAs (nat-siRNAs), heterochromatic small interfering RNAs (hc-siRNAs), trans-acting siRNAs (tasiRNAs), and repeat-associated siRNAs (rasi-RNAs) [38,39]. Based on the genomic location relative to protein-coding genes, lncRNAs consist of long intergenic ncRNAs, intron ncRNAs, antisense ncRNAs (ancRNAs), and sense ncRNAs (slncRNAs), which are potent cis-/trans-regulators to influence the transcriptional activity of their target loci [24,40]. Meanwhile, circRNAs can be divided into exonic circRNAs, intronic circRNAs, UTR circRNAs, intergenic circRNAs, and other circRNAs deriving from two or more genes [37] (Table 1).

### 2.1. The Expression and Targeting Action Mode of Small Non-Coding RNAs

In plants, *MIR* genes transcribe to primary miRNAs by RNA polymerase Ⅱ and form special hairpin structures [41]. Then the single-strand hairpin pri-miRNAs are cleaved in two steps to generate a stem–loop intermediate (precursor miRNAs) and the miRNA/miRNA* duplex by Dicer-like 1(DCL1) and Hyponastic Leaves 1 (HYL1) protein in the nucleus [42,43]. Next, the miRNA/miRNA* duplex is methylated by HUA ENHANCER 1 (HEN1) and transported to the cytoplasm by HASTY (HST) protein. Within the cytoplasm, the miRNA duplex is unwound, and the miRNA, but not miRNA*, is incorporated in the RNA-induced silencing complex (RISC). Then miRNA interacts with the complementary target mRNA and activates the catalytic RISC with the assistance of Argonaute 1 (AGO1) [44]. AGO proteins consist of PAZ (Piwi Argonaut and Zwille), MID (Middle), and PIWI (P-element-induced wimpy testis) domains to bind sRNA [45,46]. The PAZ domain can attach to the 3′ nucleotide of the guide strand [47], and the PIWI domain with RNaseH-like activity can cleave the target RNA, while the MID domain can sense the identity of the 5′ nucleotide of sRNA [48,49] (Figure 1).

Plant miRNA target sites have been found within ORFs (open reading frames), 5′UTRs (5′ Untranslated Regions), and 3′UTRs, as well as noncoding transcripts [50]. MiRNAs mainly inhibit gene expression at the post-transcriptional level through directing cleavage of mRNA targets for cleavage or repress translation with the participation of VARICOSE (VCS), AGO1, and AGO10 protein [51,52]. The transcription of miRs is determined by Cis-regulatory elements and trans-acting regulators or by epigenetic modification. In *Arabidopsis*, AGL15 and AGL18 (AGAMOUS-like proteins) form heterodimer and bind to the CArG (CC (A+T-rich) 6 GG) motifs of the MIR156 promoter to activate the MIR156 expression [53]. Moreover, SPL9 and SPL15 which are regulated by miR156 directly promote the transcription of miR172 [54]. Moreover, in *Arabidopsis*, the null mutants of *HESO1* (*HEN1 SUPPRESSOR1*), which can uridylate the unmethylated miRNAs, increased the transcription of miR166/165, miR169, miR171/170, and miR172 [55]. Numerous miRNA-targets, such as miR156/SPL [56,57], miR172/IDS1 [58], miR393/(TIR1/AFB2) [59], miR160/(ARF10/ARF16/ARF17) [60], and miR159/(MYB33/MYB65) [61], have been identified to affect plant growth, development, and abiotic stress responses through directly regulating their targeting mRNA level. Furthermore, miRNA172 targets the APETALA2 transcription factor *SlAP2a* to control the fruit development in tomato (*Solanum lycopersicum*) [62].

MiRNAs regulate plant development and abiotic-stress response through targeting functional genes transcript cleavage and translation repression [63,64]. In plants, miRNA recognizes target binding sites of mRNA through exact or nearly exact sequence complementarity that results in the cleavage at the tenth nucleotide of miRNA complementary sites [65] or translational inhibition [66]. MiRNAs also inhibit the translation of target mRNAs through partially mismatched sequences in their 3′ UTR without degrading the mRNA. Moreover, miRNAs control translation initiation by inhibiting eukaryotic initiation factor 4E/cap and poly(A) tail function [67]. The functions of miRNAs are determined by their binding regulatory targets, and the miRNA complementary sites within the mRNA targets for miRNA binding are relatively conserved and important [68]. Due to the nature of miRNA:mRNA base pairing in plants, many of the targets are members of the same gene families; however, the fraction of the gene family members with miRNA complementary sites varies considerably. In *Arabidopsis*, 10 of the 16 Squamosa-promoter Binding Protein (SBP)-like genes have miR156 complementary sites, while only 5 of over 100 *MYB* and *NAC* (*NAM*, *ATAF1/2*, and *CUC2*) family genes have sites complementary to miR159 or miR164, respectively [69]. Furthermore, the targets of the same miRNA can be different gene family members; conversely, different unrelated miRNAs can be complementary to different members of the same gene family. For example, miR319 can target not only the *MYB* family member, but also the *TCP* (*Teosinte branched1/Cycloidea/proliferating cell factors*) genes [70], while miR160, miR167, and miR390 can target different members of the Auxin Response Factor family [60,71,72]. The specific binding sites’ recognition and regulation cause miRNAs to be involved in multi-metabolic pathways. In rice, miR319 can positively regulate cold tolerance by targeting *OsPCF6* (*Proliferating cell factor 6*) and *OsTCP21* [73], and miR319-targeted *OsTCP21* and *OsGAMYB* also regulate the tillering and grain yield [70]. In addition, miR319 targets *TCPs* to regulate leaf development in *Arabidopsis* [74] and promotes the transition from cell elongation to wall thickening in cotton fiber [75]. Moreover, miR319a-targeted PtoTCP20 can regulate secondary growth via interactions with PtoWOX4 (WUSCHEL-related homeobox 4) and PtoWND6 (wood-associated NAC domain 6) in *Populus tomentosa* [76]. It is noteworthy that the specific binding site of target mRNA is not the only factor for miRNA to bind and splice in regulating the abundance of the target mRNAs. For instance, almost 100 potential miR159 targets with four or fewer mismatches can be obtained by using the standard target prediction program, psRNATarget, in *Arabidopsis* [77], but among the top twenty targets, which include eight *MYB* genes with highly conserved miR159 binding sites, only *MYB33* and *MYB65* are strongly downregulated by miR159-mediated cleavage [78]. Similarly, in *Brassica rapa*, only 4 of 13 *SPL* candidate targets with highly conserved binding sites of miR156/157 were cleaved [79], and three of seven candidate targets of miR828 were proved to be spliced by RLM-5′RACE [80]. Furthermore, the differential miR159-mediated silencing has been proved to be correlated with the differential RNA secondary structure of miR159 targets and physically spatial and/or temporal separation of miR159 and targets in plant tissues [78]. Then dynamic RNA secondary structures in vivo may be operating as a riboswitch, with certain formations facilitating silencing of specific miRNA targets [78].

Except for directly regulating the expression of targets, primary miRNAs of miR171b in *Medicago truncatula* and miR165a in *Arabidopsis thaliana* have been reported to produce the peptides miPEP171b and miPEP165a, which increase the transcription of pri-miRNA and the accumulation of miR171b and miR165a in a positive feedback loop and lead to the reduction of lateral root development and stimulation of main root growth, respectively. The pri-miRNAs possess both protein-coding and non-coding roles [81]. In soybean (Glycine max), miPEP172c also takes a positive role in miR172c accumulation, resulting in an increase in nodule formation [82]. In addition, miPEP858a in *Arabidopsis* regulated the expression of pri-miR858a and associated target genes to control plant development and the phenylpropanoid pathway [83].

Moreover, plant miRNAs can act as transmitters to transport from cell to cell and between distant organs via the long-distance to communicate with environments for metabolism, growth, reproduction, and defense reactions [84,85]. The exogenous miRNAs miR156 and miR399 have been demonstrated to repress the mRNA level of their targets via an RNAi mechanism when transferring between neighboring plants [86]. The root-derived Nb-miR164 can modify the scion trait via long-distance movement in *Arabidopsis*/*Nicotiana* interfamilial heterograft [87]. Moreover, miRNAs have also been shown to play a crucial role in extracellular vesicles to fungal pathogen to silence virulence genes and in shoot meristems and root vascular systems [88,89,90]. Additionally, miRNAs can realize communication crossing species to mediate co-evolution between species. For example, plant miRNAs enriched in beebread but not in royal jelly lead to delayed development and decreased body and ovary size in honeybees, preventing larval differentiation into queens [91]. Moreover, miR2911, identified in Traditional Chinese Medicine honeysuckle (HS), can directly target *Influenza A viruses* (*IAVs*) with a broad spectrum to suppress viral infection [92]. Furthermore, plant miRNAs have also been reported to reduce cancer-cell proliferation by targeting *MALAT1* (*Metastasis-associated lung adenocarcinoma transcript 1*) and *NEAT1* (*Nuclear-enriched abundant transcript 1*) [93]. Therefore, the “mobile” role of miRNAs from different origins under specific environments and cross-species transmission to produce “amplified” secondary siRNAs enhance the potential effect on their targets and downstream developmental pathways.

Similar to the biogenesis pathway of miRNAs, siRNAs are derived from perfectly double-stranded RNAs, which are transcribed from inverted repeats, natural cis-antisense transcript pairs, and single-stranded RNA through the action of RNA-dependent RNA polymerases (RDRs). The DCLs proteins cleave dsRNAs into 21-to-24 nt siRNAs, which are loaded into AGO protein to form RISC that guides target regulation at the posttranscriptional level or the transcriptional level through RNA-directed DNA methylation (RdDM) [39]. In plants, DICER-LIKE PROTEIN 3 (DCL3) produces 24-nucleotide (nt) small interfering RNAs (siRNAs) that determine the specificity of the RNA-directed DNA methylation pathway [94]. While the endogenous 22-nucleotide siRNAs are generated by the DCL2 protein. The 22 nt siRNAs derived from NIA1/2 (nitrate reductases) can mediate translational repression, inhibiting plant growth and enhancing stress responses [95]. Moreover, DCL4 generates 21 nt siRNAs from double-stranded RNAs responsible for degrading TE (transposable element) mRNAs in the somatic plant body, and the siRNAs have specifically been shown to participate in cell-to-cell silencing [96,97]. Not only can siRNA mediate transcriptional gene silencing through RNA-directed DNA methylation, but they can also perform post-transcriptional gene silencing through cleavage and translational inhibition without changing the DNA sequence [98,99]. While the biogenesis and regulatory role of siRNAs have been reported and reviewed in detail [33,39,98,99,100,101], the function of siRNAs involved in plant development and abiotic stress response still needs to be explored.

### 2.2. Types of Long Non-Coding RNA and Their Actions in Gene Expression

In eukaryotes, almost 90% of the genome was found to be transcribed by using high-throughput sequencing technology, but only approximately 2% corresponds to protein-coding mRNAs [3,102]. Plant lncRNAs are transcribed by RNA polymerases PolⅡ, PolⅣ, and PolⅤ, which are involved in the regulation of gene expression [6]. LncRNAs may be transcribed in a stand-alone unit from enhancers, promoters, and introns of genes; from pseudogenes; or from antisense to other genes with varying degrees of overlap [102]. According to the location relative to protein-coding genes in the genome, lncRNAs can be defined as natural antisense transcripts (lncNATs), intronic lncRNAs, intergenic lncRNAs (lincRNAs), and sense lncRNAs. LncNATs initiate inside or 3′ to a protein-coding gene and show either in concordance with the sense strand transcripts or in a discordant manner, which can mediate the transcriptional or post-transcriptional regulation of genes transcribed from sense strand [103]. Intronic lncRNAs are transcribed in either direction from an intron of a protein-coding gene, without overlapping with an exon, while sense lncRNAs share the same promoter with the protein-coding gene and transcribe from a region overlapping an exon [104]. Intergenic lncRNAs are independent transcriptional units and are located outside protein-coding genes [105]. Compared with mRNAs, lncRNA transcripts are shorter and lack many motifs, such as ORFs and Kozak consensus sequences [106]. Similar to mRNA, lncRNAs have a 5′ m^7^G cap and a 3′ poly(A) tail and are processed as mRNA mimics [107] (Figure 2A). However, lncRNAs have shown low conservation in sequences among species and have low expression levels with tissue-specific expression patterns responding to various stresses in plants [108].

In *Arabidopsis*, over 40,000 candidate lncRNAs have been identified [109], and among them, about 75% are lncNATs [103], the lincRNAs are about 15% [110], and only 2708 lincRNAs (less than 50%) were detected in transcription level by RNA-seq experiments [6]. The increasing evidence indicates that the transcription levels of lncRNAs are correlated with plant growth, development, and abiotic- and biological-stress response [111,112,113,114]. Functional analyses of lncRNAs have revealed that they act as scaffolds, guides, and decoys in *cis* or *trans* to regulate gene expression through transcription, epigenetic modification, and post-transcriptional gene regulatory mechanisms [3,115,116] (Figure 2B). As scaffolds, lncRNAs with sequences complementary to RNA or DNA can be recognized via specific sequence motifs or secondary/tertiary structures [116]. LncRNAs can also bind to histone-modifying complexes and DNA-binding proteins (including transcription factors) and control gene expression by modifying chromatin accessibility, transcription, splicing, and translation levels. In *Arabidopsis*, lncRNA HID1 (HIDDEN TREASURE 1) binds to the promoter of the *PIF3* gene to downregulate its expression and affect the photomorphogenic process [117]. In addition, the guiding lncRNAs interact with transcriptional co-regulators or chromatin regulatory protein complexes and recruit them to a specific DNA region to modulate transcription. For instance, in the early vernalization process of plants, lncRNA COLDAIR recruits PRC2 (Polycomb Repressive Complex 2) in the first intron region of *FLC* (*FLOWERING LOCUS C*) and leads to the H3K27me3 modification of histones [118]. Moreover, as decoys, lncRNAs bind miRNAs, such as ceRNAs (competing endogenous RNAs), or proteins in the nuclei to mimic and compete with their consensus DNA-binding motifs to control the functions of regulatory miRNAs or proteins [119]. For example, in the tomato, lncRNAs have been identified as ceRNAs and predicted to decoy 46 miRNAs, and among them, lncRNA42705 and lncRNA08711 show a negative correlation with the expression of miR159 and a positive correlation with the expression levels of *MYB* genes upon *Phytophthora infestans* infection [120]. Similarly, in maize, the lncRNA PILNCR1 (Pi-deficiency-induced long-noncoding RNA1) also serves as mimic to sponge miR399 and releases the *PHO2* (*Phosphate 2*) mRNA, which negatively regulates phosphate transporters [121].

LncRNAs are usually long, and their sequences are not fully complementary to the target sequences that are involved in splicing, gene inactivation, and translation [122,123] through interactions with proteins, DNA, or other RNA molecules [124]. LncRNAs take advantage of intermolecular interaction-controlling gene expression and protein activity to regulate plant resistance to biotic and abiotic stresses, flowering, and lateral root development [125]. First, lncRNAs participate in forming ribonucleoprotein complexes to modulate the subcellular localization and the molecular activity of their protein partners [126]. For example, the lncRNA ELF18-INDUCED LONG-NONCODING RNA1 (ELENA1) can recruit the transcriptional coactivator Mediator subunit 19a to activate the transcription of the *PATHOGENESIS-RELATED 1* gene to enhance the resistance against pathogenic bacteria in *Arabidopsis* [127]. Moreover, lncRNA HIDDEN TREASURE 1 (HID1) can also interact with ribonucleoprotein to bind the first intron of *PHYTOCHROME-INTERACTING FACTOR 3* and repress its transcription, promoting photomorphogenesis in continuous red light [117]. Another two lncRNAs, COLDAIR and AG-intron-4, also interact with the Polycomb repressive complex-2 (PRC2) in *Arabidopsis* to inhibit the transcription of *FLC* (*FLOWERING LOCUS C*) and *AGAMOUS*, which are involved in the regulation of flowering [118,128]. Second, lncRNAs participate in forming RNA–DNA hybrids to control the expression of neighbor genes in cis/trans or to influence chromatin conformation of target regions [126]. In *Arabidopsis*, lncRNA APOLO was identified to recognize multiple spatially unrelated loci in the genome via sequence complementarity and R-loop formation, and most of the targets are auxin-responsive genes involved in lateral root development [129]. Third, lncRNAs participate in forming RNA–RNA duplexes to regulate gene expression at the post-transcriptional level. LncNATs (natural antisense transcripts) can pair with target RNAs at specific regions of complementarity to control their stability or their translation [126]. For instance, 70% of annotated mRNAs were found to associate with detectable lncNATs in *Arabidopsis* [103]. In *Petunia hybrida*, a cis-lncNAT (the antisense transcript of the *SHO* gene) triggers the production of double-stranded RNA-derived small interfering RNAs (siRNAs), leading to the degradation of the *SHO* (*Shooting*, *Cytokinins-producing gene*) RNA in a tissue-specific manner in cytokinin regulation [130]. In *Arabidopsis*, the lncRNA INDUCED BY PHOSPHATE STARVATION 1 (IPS1) was also identified to act as target mimics for miR399 by forming a lncRNA-miRNA duplex with a mismatched loop at the miRNA cleavage site, leading to the non-cleavage of IPS1 as a miRNA sponge [131]. Thus, plant lncRNAs participate in forming ribonucleoprotein complexes, RNA–DNA hybrids, or RNA–RNA duplexes to exercise their regulation function.

## 3. Regulation Role of ncRNAs in Plant Development

### 3.1. miRNAs Involved in Plant Development

Plant development is composed of growth and differentiation, which include all changes that an organism goes through during its life cycle, from the germination of the seed to senescence. Plant development processes are determined by many proteins and phytohormones, whose expression levels could be directly or indirectly regulated by some non-coding RNAs (ncRNAs) [132]. MiRNAs are key regulators and play important roles in plant development, such as seed development and germination, shoot apical meristem (SAM) maintenance, floral organ identity, leaf morphogenesis, root initiation, and development.

#### 3.1.1. miRNAs in Seed Development and Germination

Germination depends on the seed’s physiological state (dormancy), which is determined by the balance of the ABA/GA (abscisic acid/gibberellins) ratio [133]. In *Arabidopsis*, the overexpression of miR160 caused hyposensitivity to abscisic acid (ABA) during the seed germination process [134]. ARF10 and ARF16, two of the miR160 targets, regulate *ABI3* expression to induce seed dormancy [135]. In cotton, the miR160/ARF axis also influences seed size by directly or indirectly regulating seed development-associated genes [136]. Additionally, miR159 is involved in the regulation of seed dormancy and germination because the overexpression of *MIR159* renders seeds hyposensitive to ABA and consistently suppresses transcript levels of *MYB33* and *MYB101*, which are positive regulators of ABA responses and targeted by miR159 [137]. Then there is crosstalk between ABA and auxin in seeds’ germination, and the downregulation of a component for auxin signal transduction by miR160 may be a regulatory step to decrease ABA sensitivity in mature seeds and to switch to the germination mode [138]. Moreover, miR393 can affect seed development by targeting two genes encoding the auxin receptors TIR1/AFBs in barley. Overexpression and target mimic (*MIM393*)-mediated inhibition of miR393 both affect the development of seeds [139]. Furthermore, *mir156* mutations enhance seed dormancy by suppressing the gibberellin (GA) pathway through de-repression of the miR156 target gene *Ideal Plant Architecture 1* (*IPA1*), which directly regulates multiple genes in the GA pathway [140]. The miR156-targeted SPL9 was also reported to directly activate the expression of ABA-responsive genes through binding to their promoters and to physically interact with ABSCISIC ACID INSENSITIVE 5 (ABI5) [141]. Then miR156, miR159, miR160, and miR393 are involved in the regulation of seed development and germination through the auxin, ABA, and GA pathway. Moreover, a miR164-dependent regulatory pathway, miR164-NAC32/NAC40-EXPB14/EXPB15, has also been characterized to participate in maize seed expansion [142] (Figure 3).

#### 3.1.2. miRNAs in Shoot Development and Apical Meristem (SAM) Maintenance

The shoot apical meristem (SAM) is the main plant meristem and consists of a group of dividing cells that develops to plant lateral organs, such as leaves. WUSHEL (WUS) and CLAVATA3 (CLV3) are essential regulators of meristem maintenance and differentiation, and miR394 can target and downregulate the expression of the *LEAF CURLING RESPONSIVENESS* (*LCR*) gene, which affects the WUS–CLV3 pathway in *Arabidopsis* [90]. In addition, miR394 can function as a mobile signal to move from the outer cell layer L1 (protoderm) to L3 (organizing center) layer, where WUS protein is located, restricting the expression of the target *LCR* and maintaining the shoot stem-cell niche in the SAM region [90,143]. Two other miRNAs, miR165 and miR166, are characterized to target class III *HOMEODOMAINLEUCINE ZIPPER* (*HD-ZIP III*) family genes, including *PHABULOSA* (*PHB*)*/ATHB14*, *PHAVOLUTA* (*PHV*)/*ATHB9*, *INTERFASCICULAR FIBERLESS/REVOLUTA* (*IFL1*/*REV*), *INCURVATA4*/*CORONA*/*ATHB15*, and *ATHB8*, which are involved in SAM-related development [144]. However, AGO10 can act as a decoy for miR165 and miR166 to compete with AGO1 and prevent their repressive function on *HD-ZIP III* genes and maintain the SAM development [144,145].

#### 3.1.3. miRNAs in Floral Development

A number of miRNAs and their target genes have been identified to be involved in the transition from the vegetative to reproductive phase, namely plant floral transition. In *Arabidopsis*, the overexpression of miR172 accelerates flowering through translational inhibition of floral homeotic gene *APETALA2* (*AP2*) [66,146], whereas ectopic expression of a *MIM172* mimicry constructs delays flowering [147]. MiR172 has been identified to be regulated by several members of the SQUAMOSA PROMOTER BINDING PROTEIN-LIKE (SPL) TF family [54] that are targeted by miR156 and contribute to juvenile to adult vegetative phase transition and the age-dependent pathway to flowering [148,149]. SPL9, a transcriptional activator of *MIR172* [150], also directly regulates the expression of *AP1*, *FUL*, *AGL24*, and *SOC1* through binding to their respective promoters [148]. Thus, miR172 and miR156 always display some degree of opposite correlation in expression patterns and regulatory functions. The research on the miR172 family demonstrated that, in *Arabidopsis* leaves, miR172A and miR172B are the major mediators that promote flowering under long days, while, in the SAM, miR172D takes a major role in promoting flowering under short days [146,149,151]. In contrast, the transcription level of *MIR156* gradually decreases from the seedling stage to the adult stage, and the overexpression of miR156 results in delayed floral transition [152]. In addition, miR390 represses flowering by inhibiting the activity of ARF3 and ARF4, which results in prolonging the juvenile phase [153]. MiR172-targeted AP2 represses the *ARF3* expression by directly binding to its promoter [154], while ARF3/4 regulates the expression of miR156-targeted *SPL3* [153]. Then a crosslink among miR156, miR172, and miR390 is involved in the regulation of the juvenile-to-floral transition. Moreover, miR159 targets GA-specific transcriptional regulator GAMYB-related proteins (*MYB33*, *MYB65*, and *MYB101*), which are involved in the GA-promoted activation of *LEAFY*. The overexpression of miR159 causes reduced expression of *LEAFY* (*LFY*) and delays flowering time [14]. In *Arabidopsis* and rice, miR159 is highly expressed in anthers, and its overexpression can cause male sterility, resulting from the failure of pollen release [155]. Another miRNA involved in flowering time is miR169, and the main target of miR169 is the *NUCLEAR FACTORY*, *SUBUNIT A* (*NF-YA*) TF gene family, which can bind to the promoter of *FLC* to induce its expression [156]. The miR169d overexpression in *Arabidopsis* exhibits early flowering, and, in contrast, the overexpression of the *rNF-YA2* accounts for late flowering [157]. Furthermore, miR399 plays a crucial role in regulating flowering time by downregulating the expression levels of *PHOSPHATE 2* (*PHO2*). Both miR399 overexpression and loss-of-function *pho2* mutants show early flowering phenotypes [158]. Besides miR156-SPL, miR172-AP2, miR159-MYB, miR390-ARF, miR169-NF-YA, and miR399-PHO2 participating in flowering time control, miR164-CUC1/2 (CUP-SHAPED COTYLEDON) and miR319-TCP (TEOSINTE BRANCHED1, CYCLOIDEA, AND PCF) also function in the establishment of organ boundaries during floral development [159,160,161]. The overexpression of miR164 or loss of function of its target *CUC1/2* leads to fused sepals and stamens and the loss of petals [162]. Meanwhile, the overexpression of miR319 in *Arabidopsis* causes stamen and male sterility defects, whereas the miR319 loss-of-function mutant shows narrower petals and defective anther development [163]. These miRNA-target pathways indicate that a complex network mediated by different plant hormones and development transcription factors is involved in the plant floral development (Figure 4).

#### 3.1.4. miRNAs in Leaf Development

Plant leaf initiated from an undifferentiated cell in SAM peripheral region is the major photosynthetic organ and plays a dominant role in plant biomass production. MiR164 and their targets *CUC1* and *CUC2* have been demonstrated to regulate organ boundaries in leaf development in *Arabidopsis* [153]. In strawberries, *FveCUC2a* targeted by FvemiR164a regulates leaf serrations, because the overexpression of *FveMIR164A* produces simple leaves with smooth margins, resembling the leaf phenotype of *fvecuc2a* [164]. Additionally, single mutations in the *MIR319A* or *MIR319B* gene inhibit the formation of leaf serrations in *Arabidopsis*, and double mutations of *MIR319A* and *MIR319B* increase the extent of this inhibition and result in the formation of smooth leaves. However, gain-of-function mutations in the *TCP4* gene can impair the cotyledon boundary and leaf serration formation [165]. Moreover, miR396-targeted AtGRF transcription factors regulate cell division and differentiation during leaf development in *Arabidopsis* [166]. In lettuce, smaller leaves were also observed in LsamiR396a overexpression lines, in which *LsaGRF5* was downregulated, while overexpressing *LsaGRF5* exhibited larger leaves [167]. For another, miR393 regulates the expression of the *TIR1*/*AFB2* auxin receptor involved in auxin-related development of *Arabidopsis* leaves [168]. In rice, overexpression of miR393 also leads to an enlarged flag leaf, which is related to the auxin signaling regulated by target *TIR1* homolog [169]. Furthermore, in *Arabidopsis*, miR165/166 negatively regulates *HD*-*ZIP IIIs* (*Class-III homeodomain-leucine zipper*) to maintain the abaxial–adaxial polarity of the leaf [170,171]. Moreover, SlymiR208 is identified as a positive regulator in leaf senescence through negatively regulating CK (Cytokinin) biosynthesis via targeting *SlIPT2* and *SlIPT4* (Isopentenyltransferases) in tomato [172].

#### 3.1.5. miRNAs in Root Development

Roots are essential for plant fixation, nutrient, and water uptake, and several miRNAs have been identified to regulate root growth and patterning by targeting different transcription factors or genes [173]. *NAC1* targeted by miR164 has been demonstrated to provide a homeostatic mechanism to a downregulated auxin signal for lateral root development in *Arabidopsis* [174]. Nevertheless, the formation of adventitious roots is also regulated by auxin-related miRNAs through various ARF transcription factors [175]. Thus, *ARF6* and *ARF8* transcripts targeted by miR167 serve as negative regulators of adventitious root development in *Arabidopsis* [176]. In addition, crown root development in rice is regulated by CRD1 (Crown Root Defect 1) via the CRD1-miR156-SPL pathway [177]. In *MIR156OE Arabidopsis* plants, *SPL10* is significantly downregulated, and the expression of *AGL79* (*AGAMOUS-like MADS-box protein 79*), which is directly regulated by SPL10, is also reduced to repress lateral root growth. This suggests a role for the miR156-SPL10-AGL79 module in plant lateral root growth [178]. Moreover, the NF-YA2 TF (transcription factor) targeted by miR169 has been recognized to control the root architecture, and loss-of-function miR169 can lead to improper root initiation [179]. Furthermore, in tomatoes, the miR171-GRAS module participates in a series of developmental processes, including root length, through modulating gibberellin and auxin signaling [180]. In grapevine (*Vitis vinifera*), the small peptide vvi-miPEP171d1 encoded by primary-miR171d can increase the transcription of *vvi-MIR171d* and promote adventitious root development. Similarly, in *Arabidopsis*, ath-miPEP171c inhibits the growth of primary roots and induced the early initiation of lateral and adventitious roots, but the exogenous application of vvi-miPEP171d1 cannot result in any phenotypic changes in *Arabidopsis*, and ath-miPEP171c has no effect on grape root development [181]. Thus, miPEP171d1 regulates root development by promoting *vvi-MIR171d* expression in a species-specific manner [181]. Additionally, in apples, *mdm-MIR393b* is involved in mediating auxin signaling and inducing adventitious root formation by targeted regulation of *MdTIR1A* expression in apple rootstock [182]. In rice, miR393 has also been identified to negatively regulate the miR390-mediated growth of lateral roots under stress [183]. Another miRNA that regulates auxin-related pathways in root development is miR847, and miR847 targets *IAA28* (an IAA/ARF transcriptional repressor), which can interact with ARF proteins and promote lateral root formation [184,185]. Then miRNAs involved in multi-signaling pathways regulate plant root development (Figure 5).

#### 3.1.6. miRNAs in Shaping the Fruit/Grain Size and Maturation

Fruits and seeds are important productions for plants to propagate offspring, and they also supply crop yields for global creatures. Phytohormones take important roles in shaping the fruit/grain size and maturation. NcRNAs have been reported to be involved in the regulation of the phytohormones-mediated pathways. For instance, auxin-GA crosstalk via ARFs can be regulated by miR160 and miR167 to influence the duration of fruit development by affecting ABA accumulation [186]. For fruit ripening, ethylene plays a clear role in climacteric fruits, such as the tomato, whereas non-climacteric ripening fruits, such as the strawberry, are generally associated with ABA. Moreover, NAC transcription factors respond to ABA and ethylene [187,188] and can be targeted by miR164 to control the ripening of the strawberry [189] and kiwifruit [190]. Additionally, in the tomato, SlymiR157 has been reported to affect *LeSPL*-*CNR* (*SQUAMOSA Promoter Binding Protein-like*) expression and influence fruit ripening [191]. The level of the intact messenger of *AP2a* (*APETALA2a*) is also actively modulated by miR172 during the fruit ripening of the tomato [192]. Furthermore, the overexpression of miRNA172p reduces the fruit size of apple “Royal Gala”, while reduced expression of miRNA172 by a transposon insertion associates with large fruit. The expression patterns of miRNA172p and *AP2* genes in fruit and the target site of miRNA172 in *AP2* genes indicate that miRNA172p could modulate levels of AP2 proteins to influence apple fruit development [193]. In strawberry, miR159 is also identified to target and regulate the expression of *Fa-GAMYB* during berry receptacle development and cooperatively changed GA endogenous levels [194]. Another miRNA, miR397, could increase grain size and promote panicle branching when OsmiR397 is overexpressed and downregulates its target *OsLAC*, which is involved in the sensitivity of rice plants to brassinosteroids [195]. Additionally, in rice, OsPIL15 (phytochrome-interacting factor-like 15) also activates the expression of *OsMIR530*, which targets *OsPL3* (PLUS3 domain-containing protein) to regulate grain yield. Overexpressed *OsMIR530* or knocking out *OsPL3* decreases the rice yield by altering the grain size and panicle architecture, whereas blocking OsmiR530 increases grain yield [196]. Then miR160, miR167, miR164, miR157, miR172, miR159, and miR397 are all involved in the regulation of the fruit/grain size and maturation through the phytohormone pathway; however, OsmiR530 has not been identified to be related to phytohormones.

### 3.2. LncRNAs Involved in Plant Development

In plants, lncRNAs have been shown to participate in the regulation of developmental processes (Table 2). The expression of the major flowering repressor FLOWERING LOCUS C (FLC) involved in vernalization is tightly fine-tuned by lncRNAs, such as COOLAIR, COLDAIR, ANTISENSE LONG (ASL), and COLDWRAP. COOLAIR is natural antisense transcript originating from the 3′end of the *FLC* locus and represses *FLC* through increasing the level of histone demethylase FLD, leading to H3K4me2 demethylation of *FLC* [197,198]. Additionally, COLDAIR transcribed from the second *FLC* intron can bind PRC2 complex protein CURLY LEAF (CLF) to recruit PRC2 to the *FLC* locus allowing deposition of the repressive H3K27me3 chromatin mark to repress *FLC* [118,199]. Moreover, COLDWRAP is associated with the promoter of *FLC*, which also interacts with CLF to form an intragenic chromatin loop and to confer *FLC* repression [128]. However, ASL, a non-polyadenylated antisense transcript with an unknown function, is also transcribed from the *FLC* locus [200]. Another natural antisense lncRNA, MAS, originated from the *MADS AFFECTING FLOWERING4* (*MAF4*) locus also involved in vernalization and regulates *MAF4* via interacting with histone-modifying enzyme WDR5a (WD40-REPEAT 5a) [201].

Apart from regulating vernalization, LncRNAs take important roles in floral organ identity and flowering time control. Through RNA-seq technology, the long intergenic noncoding RNAs LINC-AP2 has been identified to regulate the floral structure of shorter stamen filaments by anti-cis downregulating AP2 gene in TCV-infected *Arabidopsis* plants [202]. In addition, bra-eTM160-1 and bra-eTM160-2 are also identified to be functional target mimics for miR160 through the miR160-*ARF* pathway regulating pollen development and fertilization in *Brassica rapa* [25]. Other lncRNAs such as FLOWERING LONG INTERGENIC NON CODING RNA (FLINC), CDF5 LONG NONCODING RNA (FLORE), LONG-DAY SPECIFIC MALE-FERTILITY-ASSOCIATED RNA (LDMAR), PHOTOPERIOD-SENSITIVE GENIC MALE STERILITY 1 (PMS1T), and EARLY FLOWERING-COMPLETELY DOMINANT (Ef-cd) are flowering time-related lncRNAs. FLINC has been reported to regulate *FT* (*FLOWERING LOCUS T*) expression involved in flowering [203]. In addition, FLORE, the antisense to *CDF5* (*CYCLING DOF FACTOR 5*), can improve photoperiodic flowering by repressing the transcription of *CDFs* and increasing FT activity to promote flowering [204]. Whereas in rice, sufficient transcripts of LDMAR are required for normal pollen development of plants grown under long-day conditions, but the change of secondary structure of LDMAR due to a single nucleotide polymorphism (SNP) mutation can increase methylation in the promoter of LDMAR and reduced the transcription of LDMAR, resulting in photoperiod-sensitive male sterility (PSMS) [205]. Similarly, PMS1T also contributes to photoperiod-sensitive male sterility by producing phase-siRNAs in a miR2118-dependent manner under long-day conditions [206]. However, Ef-cd originated from the antisense strand of the flowering activator *OsSOC1* locus, recruits SDG724 complex to increase the H3K36me3 level in the *OsSOC1* locus, and positively regulates the expression of *OsSOC1*, reducing the time-span for plant maturity [207]. Besides, in rice, the parent-of-origin lncRNA MISSEN has also been proved to function by hijacking a helicase family protein (HeFP), leading to abnormal cytoskeleton polymerization during endosperm development [208]. Moreover, circRNAs derived from exon 6 of the *SEPALLATA3* (*SEP3*) gene increase the abundance of the cognate exon-skipped AS variant (SEP3.3 which lacks exon 6) through R-loop formation, in turn driving floral homeotic phenotypes of *Arabidopsis* [209].

Furthermore, for leaf development, the plant photomorphogenesis-related lncRNA, HIDDEN TREASURE 1 (HID1), modulates the chromatin structure in the *PHYTOCHROME INTERACTING FACTOR 3* (*PIF3*) promoter and represses the transcriptional activity of *PIF3*, inhibiting hypocotyl elongation of *Arabidopsis* seedlings [117]. Additionally, the AG-incRNA4 can associate with CLF to repress *AG* expression in leaf tissues through H3K27me3-mediated repression and to autoregulate its expression level [210]. Moreover, the antisense long noncoding RNA, TWISTED LEAF, maintains leaf-blade flattening by mediating chromatin modifications of *OsMYB60* and suppressing its expression in rice [211].

Another lncRNA, AUXIN-REGULATED PROMOTER LOOP (APOLO), regulated by ARF7 participates in the genetic regulatory network governing lateral root development through polar auxin transport by controlling chromatin loop dynamics [212]. APOLO regulates two homolog genes, namely *serine/threonine protein kinases PINOID (PID)* and *WAG2*, by affecting local chromatin loop formation [129]. Moreover, APOLO interacts with the transcription factor WRKY42 and directly recognizes the locus encoding the root hair (RH) master regulator ROOT HAIR DEFECTIVE 6 (RHD6), modulating its transcriptional activation and leading to low-temperature-induced RH elongation [213]. Even further, APOLO and WRKY42 can positively control the expression of several cell wall EXTENSIN (EXT) encoding genes, including *EXT3*, regulating RH development and growth [214].

Moreover, a lncRNA LAIR transcribed from the antisense strand of the neighboring gene *LRK* (*leucine-rich repeat receptor kinase*) cluster can regulate the neighboring gene cluster expression in rice. LAIR overexpression increases rice grain yield and upregulates the expression of several *LRK* genes through binding histone modification proteins OsMOF and OsWDR5 to enrich H3K4me3 and H4K16ac at the activated *LRK1* genomic region and the 5′ and 3′ untranslated regions of *LRK1* gene [215].

## 4. Function of ncRNAs in Plant Abiotic Stress Response

### 4.1. miRNAs Play Important Roles in Heat and Cold Stress

Plants are constantly exposed to various environmental stresses, such as extreme temperatures, high salinity, and drought. These abiotic stresses significantly affect plant growth and productivity. Then plant miRNAs involved in a variety of stress responses also play essential roles in plant development.

Heat is one of the most serious stresses affecting plant growth, development, and crop yields. Heat shock proteins (HSPs) and heat stress transcription factors (HSFs) have been identified to be involved in responses to heat stress in plants [216]. In *Arabidopsis*, heat stress induces the transcription of miR160, and represses the expression of *ARF10*, *16*, and *17*, regulating the high expression level of *HSP* genes, resulting in the thermotolerance of plants [217]. MiR393 is also involved in auxin-related development in plants, and overexpression of osa-miR393a in transgenic creeping bentgrass increases heat tolerance by repressing its targets *AsAFB2* and *AsTIR1* and inducing expression of *HSPs* [218]. Additionally, TamiR159-targeted cleavage of *TaGAMYB* regulates anther development and heat response possibly through the GAMYB-amylase pathway for starch degradation in wheat [219]. MiR319, with a high degree of sequence identity to miR159, targeting *TCP* and *MYB* genes, can regulate transcription levels of heat-stress-responsive genes and conferred heat stress tolerance in miR319-overexpressed *Solanum Lycopersicum* [220]. Similarly, in *Arabidopsis*, during recovery from heat stress (HS), the miR156-SPL module sustains the expression of HS-responsive genes (e.g., *HSFA2* and *HSPs*) and mediates the response to recurring heat stress, and thus may also integrate stress responses with development [221]. Conversely, miR398 is rapidly induced in response to HS, downregulating its target genes (*CSD1/2*, *copper/zinc superoxide dismutase1/2*; *CCS*, *copper chaperone for superoxide dismutase*) and resulting in ROS (reactive oxygen species) accumulation and increased HSF and HSP levels, which can regulate miR398 with a regulatory loop again [222]. Moreover, miR396 downregulates *HaWRKY6* during early responses to high temperature in sunflower, and heterologous *Arabidopsis* plants expressing a miR396-resistant *HaWRKY6* gene exhibited sensibility to high-temperature damage [223]. Furthermore, in rice, both miR166 and miR169 can target *SGT1* (the G2 allele of *skp1*), which could bind to HSP90 and HSP70 and regulate the rice response to thermal stress during flowering [224]. Another osa-miR5144 has also been identified to be involved in HS tolerance of rice by mediating the expression of *OsPDIL1;1* (*protein disulfide isomerase*), which regulates the formation of protein disulfide bonds [225]. For the *MIR400* family, a heat-stress-induced alternative splicing in the intron of *MIR400*, increasing accumulation of miR400 primary transcripts and reducing the level of mature miR400, leads to miR400 acting as a negative regulator in plant heat-stress resistance [226]. Therefore, the hormone/development/alternative splicing/ROS/HSPs and HSFs-related miRNAs are involved in heat stress responses in plants (Figure 6).

Continuous cold stress affects plant growth and development by altering cell structure and physiological and biochemical metabolism [227,228]. During cold stress, *ICE* (Inducer of CBF expression) is activated by ABA-independent pathway which activates downstream transcription factor CBF/DREB1 that binds to C-repeat elements (CRT)/low-temperature-response elements (LTRE) and induces the expression of cold-responsive (*COR*) genes [229,230]. Numerous miRNAs have been reported to play an active role during cold stress in plants. In rice, the expression of Osa-miR319b is downregulated by cold stress, but the overexpression of Osa-miR319b causes downregulation of its targets *OsPCF6* and *OsTCP21* and upregulating cold stress-responsive genes such as *DREB1*/*CBF* (*Dehydration Responsive Element Binding Protein/C-repeat Binding Factor*), *DREB2A*, and *TPP1/2* (*Trehalose-6-phosphate phosphatase 1/2*), leading to enhanced tolerance to cold stress [73]. Additionally, genetically downregulating the expression of miR319-targeted genes, *OsPCF5* and *OsPCF8*, also results in enhanced cold tolerance after chilling acclimation in rice [231]. Moreover, *OsWRKY71* can be positively regulated by the target of OsmiR156, OsSPL3, and OsWRKY71 negatively regulate the transcription factors *OsMYB2* and *OsMYB3R-2*, which counteracts cold stress by activating the expression of the stress-response genes *OsLEA3*, *OsDREB2A*, *OsCTP1*, etc. [232]. Furthermore, overexpressing miR394 in *Arabidopsis* downregulates the expression of its target *LCR* (*LEAF CURLING RESPONSIVENESS*) and activates cold-responsive genes *CBF1*, *CBF2*, and *CBF3* to be involved in plant cold tolerance [233,234]. Under cold stress, transcription factors involved in the auxin metabolic pathway play an important role. The miRNA-target pair miR169/*NF-YA* module in *Arabidopsis* have been identified to function in the Aux/IAA14-mediated cold stress response [235]. Another miR167 and tasiRNA-ARF in wheat also play roles in regulating the auxin-signaling pathway and possibly in the developmental response to cold stress [236]. For miR408 and miR397, they both target different members of laccases involved in low-temperature stress responses. MiR408 primarily targets the phytocyanin family of proteins and laccases, and miR408-OE *Arabidopsis* lines are reported to exhibit enhanced LT tolerance by modulating ROS homeostasis and lignin biosynthetic pathway [237,238]. Whereas, miR397 targets three laccases (*LAC2*, *LAC17*, and *LAC4*) and a casein kinase β subunit 3 and miR397-OE plants have been shown an increased freezing tolerance in *Arabidopsis* through the lignin biosynthesis pathway [237,239] (Figure 6).

### 4.2. miRNAs Mediate Salinity Stress Tolerance

Salinity, as a major environmental stress, affects plant growth and development. The regulatory roles of plant miRNAs and their target genes under salt stress have been gradually revealed. The expression levels of miRNA are up- or downregulated by salinity stress. The expression of miR160 and its *ARF* target gene is induced by salt stress in peanuts, and the miR160 overexpression mediated ARF18 pathway protects seedling development against the effects of ROS under salt stress [240]. In addition, overexpressing miR156a weakens salt resistance in apples, whereas its target gene, *MdSPL13*, strengthens salt resistance [241]. Moreover, the same miRNA or different miRNA from the same miRNA family have different promotion or inhibition effects on salt tolerance in different plants. For instance, overexpressing Osa-miR393 in rice and *Arabidopsis* reduces tolerance to salt [242,243], and the overexpressing miR393-resistant form *mTIR1* in *Arabidopsis* enhances salt tolerance [244]. Conversely, overexpressing Osa-miR393a in creeping bentgrass increases the uptake of potassium and improves salt-stress tolerance [218]. Similarly, overexpressing Osa-miR396c, which targets *GRF*, decreases salt and alkali stress tolerance in rice and *Arabidopsis* [245] but enhances salt tolerance in transgenic creeping bentgrass through salinity overly sensitive 1 (AsSOS1)-mediated Na^+^ exclusion [246]. Furthermore, overexpression of *PeNAC070* targeted by miR164 in *Arabidopsis* increases sensitivity to salt stresses [247], and so does overexpressing the target gene *GmNFYA3* of miR169 [248]. Another miR414c and its target *GhFSD1* (*Iron Superoxide Dismutase 1*) are also involved in the salt tolerance of cotton. Overexpressing miR414c can decrease the expression of *GhFSD1* and increase sensitivity to salinity stress by regulating reactive oxygen species metabolism [249]. Likewise, constitutively expressing miRNVL5, which targets *GhCHR* (*Cys/His-rich DC1/PHD domains*), displays hypersensitivity to salt stress by repressing *CBF* (*C-repeat binding factor*), *ERF* (*Ethylene-responsive element binding factor*), etc. [250] (Figure 7).

### 4.3. miRNAs Are Involved in Drought Stress Response

Drought stress is one of the major natural challenges that restricts the growth, development, and yield of plants. Many miRNAs participate in drought stress response in plants via auxin signaling, ABA-mediated regulation, and scavenging of antioxidants [251,252]. In apple trees, Mdm-miR160 can move from the scion to the rootstock in apples and tomatoes (*Solanum Lycopersicum*) to improve root development and drought tolerance of the rootstock. Additionally, *MdARF17* targeted by miR160 can interact with MdHYL1 (HYPONASTIC LEAVES 1) and bind the promoter of *MdHYL1* and MIR160 to activate their expression, forming a positive feedback loop regulation [253]. However, miR165/166-mediated *HD-ZIP IIIs* regulation confers drought tolerance through ABA signaling in *Arabidopsis*, while HD-ZIP IIIs activate the expression of *ARF*, which is targeted by miR160; thus, drought tolerance is mediated by miR160 and miR165/166 interactions. This interaction also triggers differential expressions of IAA- and ABA-signaling-related genes to drought tolerance of *Arabidopsis* [254]. Moreover, both the STTM166 plants and overexpressing a miR166-resistant form of *OsHB4* (*HOMEODOMAIN CONTAINING PROTEIN4*) plants show high drought tolerance in rice, due to reduced stomatal conductance and transpiration rates [255]. Another two rice auxin receptor gene homologs (*OsTIR1* and *OsAFB2*) are downregulated in OsmiR393-overexpressing rice and reduce tolerance to drought stress and hyposensitivity to auxin [243]. Furthermore, in miR156OE alfalfa plants, the reduced expression of miR156-targeted *SPL13* increases WD40-1 to fine-tune *DFR* expression for enhanced anthocyanin biosynthesis, improving drought tolerance [256,257]. Apart from the above miRNAs and targets, the miR159-*MYB* module and miR169-*NFYA* module also participate in an ABA-dependent pathway to regulate drought responses in plants [258]. In *Arabidopsis*, miR159 accumulates and mediates cleavage of *MYB33* and *MYB101* in response to ABA and drought during seed germination [137]. Similarly, in the tomato, sly-miR159 targeting of *SlMYB33* transcription factor transcript correlated with the accumulation of the osmoprotective compounds proline and putrescine to promote drought tolerance [259]. However, miR169 is downregulated by drought stress through an ABA-dependent pathway. The *nfya5* knockout and overexpressing miR169a *Arabidopsis* show sensitivity to drought stress, while overexpressing *NFYA5* displays tolerance to drought stress [260]. Likewise, gma-miR169c targeted AtNFYA1 and AtNFYA5 in soybean, and miR169n targeted *NF-γAδ* in *Brassica napus*, negatively regulating the drought-stress response by inhibiting the transcript levels of the stress response genes [261,262] (Figure 8).

### 4.4. LncRNAs Are Involved in Plant Abiotic Stress Response

LncRNAs affect the abiotic stress response by recruiting complex mechanisms based on eTM, antisense transcription-mediated modulation, chromatin modulation, or directly regulating the transcription of various abiotic-responsive genes. In *Brassica rapa*, lncRNA (TCONS_00048391) has been identified to be an eTM of bra-miR164a and could be a ceRNA for the target gene (*NAC1*, Bra030820) of miR164a involved in heat tolerance in Chinese cabbage [263]. Moreover, in cassava, the expression of miR164-targeted *NAC* (*NAM*, *ATAF1/2*, and *CUC2*) genes greatly decreases due to the cold-repressive lincRNA159, which is the miRNA164 target mimics, and upregulates the expression of miR164 under cold treatment [264]. Conversely, the expression of lincRNA340 is induced by drought stress, accompanied by an increase of miR169-targeted *NUCLEAR FACTOR Y* (*NF-Y*) genes after drought treatment [264]. Additionally, the lncRNA DROUGHT-INDUCED LNCRNA (DRIR) in *Arabidopsis* positively regulates ABA-mediated drought and salt-stress responses. Plants overexpressing DRIR display enhanced salt and drought tolerance through functioning at or upstream of the stage of gene transcription in the stress or ABA signal transduction pathways [265]. Furthermore, in cotton, lncRNA973 affects miR399 and its target *PHO2* expression involved in response to salt stress, and lncRNA973 can also modulate the expression of reactive-oxygen-species-scavenging genes, transcription factors, and genes involved in salt-stress-related processes [30]. However, lncRNA354 functions as a competing endogenous RNA of miR160b to regulate *GhARF17/18* genes, modulating auxin signaling in response to salt stress in cotton [266]. Another LncRNA XH123 of cotton is involved in the tolerance of cold stress. The XH123-silenced plants are sensitive to cold stress, which displays chloroplast damage and increases the endogenous levels of reactive oxygen species [267]. In addition, transcriptional read-through of the lncRNA SVALKA results in the expression of a cryptic antisense *CBF1* lncRNA (asCBF1), which suppresses the expression of *CBF1* and decreases tolerance to freezing temperatures of *Arabidopsis* [268]. Moreover, lncRNA APOLO, which interacts with the TF WRKY42, activates *RHD6* (*ROOT HAIR DEFECTIVE 6*) transcription by modulating the formation of a local chromatin loop encompassing its promoter region to induce RH growth in response to low temperatures [213].

Despite a great number of lncRNAs having been obtained through RNA-seq, the biological role and mechanisms of action in abiotic stress response remain poorly understood. With the increasing availability of reference genome sequences and the development of molecular biology in plants, the function of various lncRNAs involved in abiotic stress response will be explored by comparative genome analysis and genetic transformation analysis.

## 5. Conclusions and Perspectives

NcRNAs do not function directly in plant growth and development or in plant response to environmental stress. Instead, ncRNAs are involved in plant development and response to abiotic stresses through regulating key components of complex gene networks, including various phytohormone signaling pathways, transcription factors (TFs), and metabolism-related genes. Previous research has indicated that ncRNAs respond to development and environmental stresses in an ncRNA-, stress-, tissue-, and genotype-dependent manner [116,132,269]. Moreover, even to the extent that the same ncRNA displays different expression trends in different plant species, and the opposite effect on the same abiotic stress in different plants. The possible reason is related to the multi-targets of ncRNA, resulting in multi-functions. For instance, miR858 inhibits the expression of *MYBL2* and positively regulates anthocyanin biosynthesis in *Arabidopsis* seedlings [270], whereas miR858 negatively regulates anthocyanin biosynthesis by repressing *AaMYBC1* expression in kiwifruit [271]. Moreover, different ncRNAs with distinct types of targets can interact and co-participate in the same or different metabolic pathway regulation, and the expression of these ncRNAs and their targets is upregulated or downregulated to be involved in positive or negative regulation in one metabolic pathway. The further research field is how the development and stress signals are sensed to promote the expression of ncRNAs and transduced to specially regulate the transcription of target genes in different metabolic pathways. Then the functions of ncRNAs in various plant species, organs, and tissues, and in response to different abiotic stress, need to be further explored with functional genomics experiments.

The “C-value paradox” shows the no-obvious correlation between DNA amount [272] and organism complexity, and the discovery that much of the genome does not encode protein-coding genes gives the paradox an “explanation”. Moreover, the functional research of non-coding RNA indicates that the termed “junk DNA” of noncoding space in a genome can be transcribed and has important regulatory roles in plant development and environmental response. Then the central role of ncRNAs in the plant constitutes a major characteristic feature of the plant kingdom, although the characteristics and function of the majority of ncRNAs are currently not known. Unraveling the complexity, biogenesis, and action of plant ncRNAs remains an important challenge. We must also keep in mind that the genome of plants has been evolved to be a sophisticated and accurate regulation system by natural selection, and the unilateral explanation to gene expression regulation from structure, function, and mechanism is not adequate. Functional proteins, transcription factors, ncRNAs, small peptides, and epigenetic modification interact to form complex and fine regulating networks of gene expression during plant development or stress responses. Therefore, perspective research in the role of ncRNAs in plants needs to focus on functional analysis and on developing transgenic abiotic-stress-resistant, top-quality, and high-yield crop plants.

## Figures and Tables

**Figure 1 ijms-23-04124-f001:**
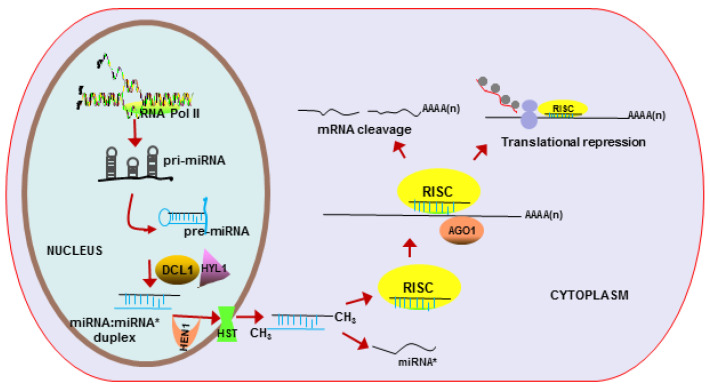
Transcription of miRNA and its regulatory role in plant cell. miRNAs are transcribed by RNA PolyII to produce pri-miRNAs (hairpin structures) in the nucleus that are processed to methylated miRNA/miRNA* duplex and are then transported to the cytoplasm to form the RISC complex. The complex can either inhibit the translation or degrade the mRNA target, depending on incomplete or complete complementarity to the target mRNA sequence.

**Figure 2 ijms-23-04124-f002:**
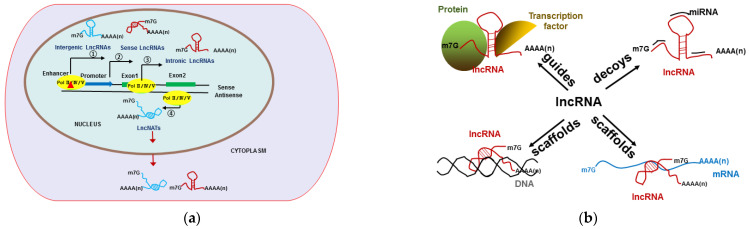
Transcription of lncRNA and its regulatory role in plant cell. (**a**) Plant lncRNAs are transcribed by RNA polymerases PolⅡ, PolⅣ, and PolⅤ from ① enhancers, ② promoters, ③ introns of genes, or ④ antisense of genes. (**b**) The lncRNAs act as scaffolds, guides, and decoys to regulate gene expression through interacting with DNA, RNA, protein and miRNAs.

**Figure 3 ijms-23-04124-f003:**
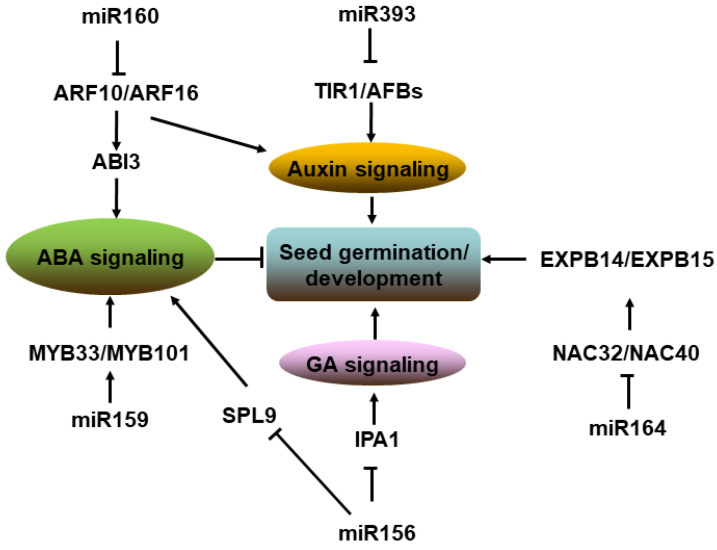
Pathway of miRNAs involved in seed germination and development. MiR156, miR159, miR160, and miR393 targeted transcription factors regulate seed development and germination through auxin, ABA, and GA pathway. Note: → represents positive regulation, and ┤ represents negative regulation.

**Figure 4 ijms-23-04124-f004:**
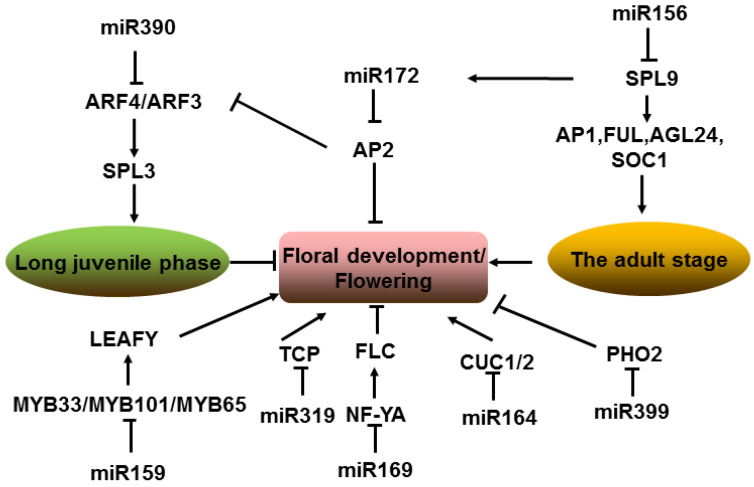
Pathway of miRNAs involved in plant floral development. MiR156, miR172, miR390, and their targets interact to form a complex network mediated by transcription factors. Note: → represents positive regulation, and ┤ represents negative regulation.

**Figure 5 ijms-23-04124-f005:**
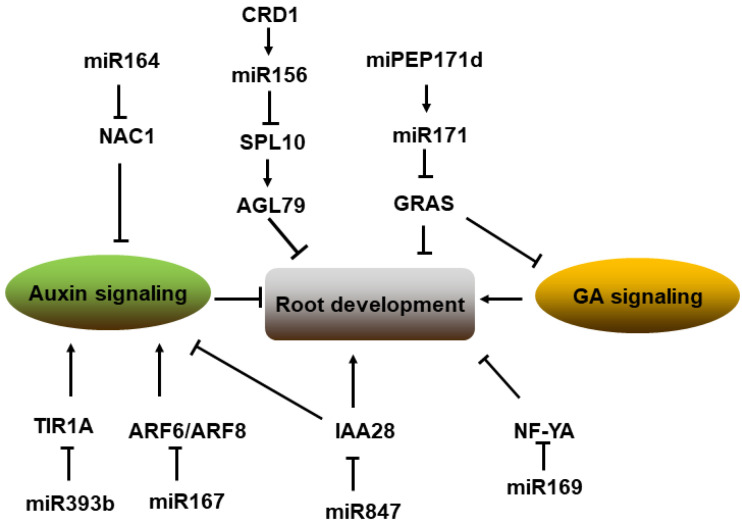
Pathway of miRNAs involved in root development of plant. MiR164, miR167, miR393, and miR171 targeted transcription factors regulate root development through auxin and the GA signaling pathway. MiR156, miR847, and miR169 also participate in regulating root development by targeting special transcription factors. Note: → represents positive regulation, and ┤ represents negative regulation.

**Figure 6 ijms-23-04124-f006:**
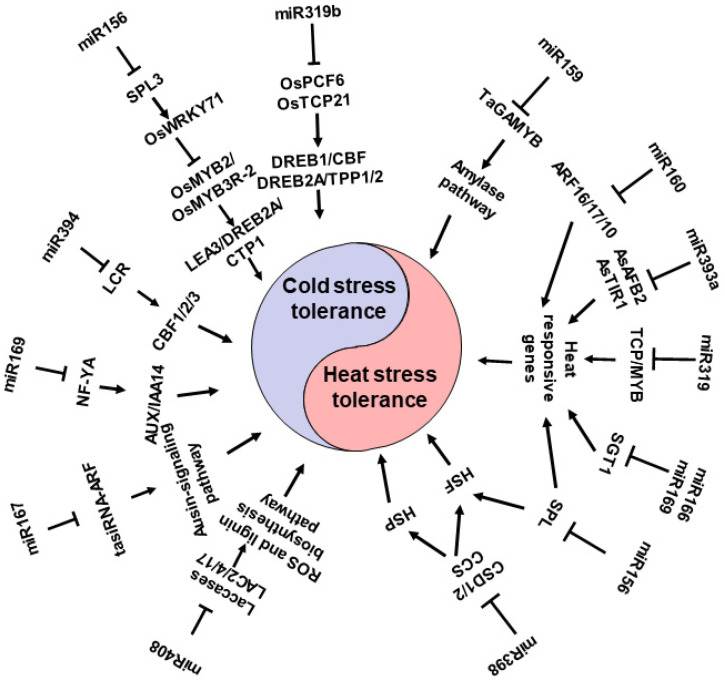
Pathway of miRNAs involved in abnormal temperature stress response in plants. MiR160, miR393, miR159, miR319, miR156, miR398, miR166/169 regulate heat stress responses in plants through the hormone/development/alternative splicing/ROS/HSPs and HSFs-related pathways. MiR319b, miR156, miR394, miR169, miR167, and miR408 also regulate cold stress responses in plants with their targets.

**Figure 7 ijms-23-04124-f007:**
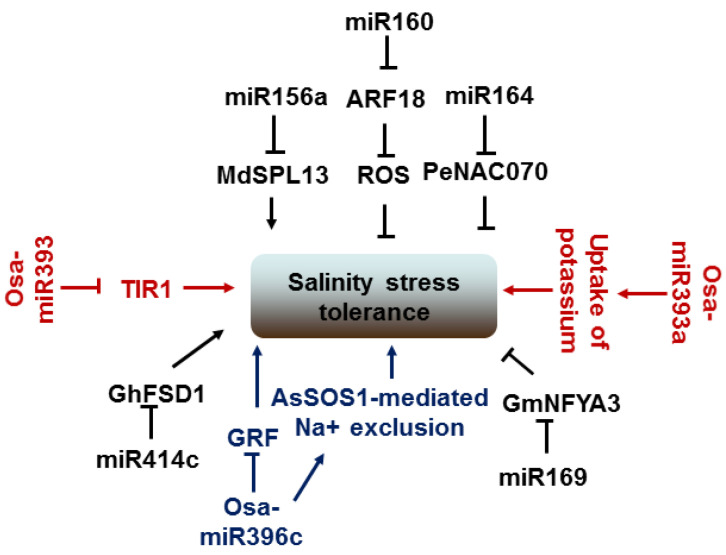
Pathway of miRNAs involved in salinity stress response. MiR156-*SPL13*, miR160-*ARF18*, miR164-*NAC070*, miR169-*NFYA3*, and miR414c-*FSD1* modules regulate plant salinity stress tolerance. MiR393 and miR396c can positively or negatively regulate salinity stress response of plant through different pathways.

**Figure 8 ijms-23-04124-f008:**
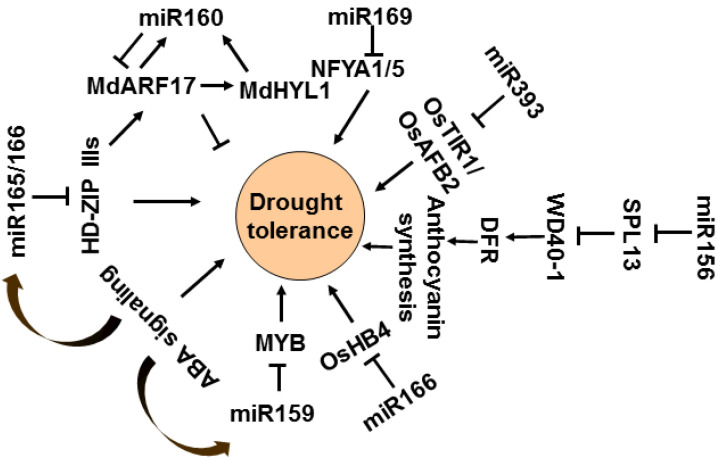
Pathway of miRNAs involved in drought stress response. MiR159, miR165/166, miR160, miR169, miR393, and miR156 participate in drought stress response in plants via auxin signaling, ABA-mediated regulation, and scavenging of antioxidants.

**Table 1 ijms-23-04124-t001:** Classification and composition of non-coding RNAs.

Classification			Composition				
Circular ncRNAs			Exonic circRNAs	Intronic circRNAs	UTR circRNAs	Intergenic circRNAs	Other circRNAs
Linear ncRNAs	Housekeeping ncRNAs		Ribosomal RNA (rRNA)	Transfer RNA (tRNA)	Small nuclear RNA (snRNA)	Small nucleolar RNA (snoRNA)	
	Regulatory ncRNAs	sRNAs	MicroRNAs (miRNAs)	Natural antisense transcript-derived small interfering RNAs (nat-siRNAs)	Trans-acting siRNAs (tasiRNAs)	Repeat-associated siRNAs (rasi-RNAs)	Heterochromatic small interfering RNAs (hc-siRNAs)
		Long ncRNAs	Intergenic ncRNAs	Intron ncRNAs	Sense ncRNAs (slncRNAs)	Antisense ncRNAs (ancRNAs)	

**Table 2 ijms-23-04124-t002:** List of plant lncRNAs associated with plant development.

LncRNAs	Targeted Gene/Protein	Description and Function	References
COOLAIR	FLD, FLC	Leading to H3K4me2 demethylation of FLC involved in vernalization and flowering.	[197,198]
COLDAIR	CLF, PRC2, FLC	Recruiting PRC2 to the FLC locus to add H3K27me3, repressing FLC to regulate flowering.	[118,199]
ANTISENSE LONG (ASL)	unknown	A non-polyadenylated antisense transcript from FLC locus, with unknown function.	[200]
COLDWRAP	CLF	Forming an intragenic chromatin loop to confer FLC repression and regulate flowering.	[128]
MAS	WDR5a	Regulating MAF4 involved in vernalization.	[201]
LINC-AP2	AP2	Regulating the floral structure of shorter stamen filaments.	[202]
Bra-eTM160-1/2	MiR160	Regulating pollen development and fertilization through the miR160-ARF pathway.	[25]
FLINC	FT	Regulating flowering.	[203]
FLORE	CDFs, FT	Regulating flowering.	[204]
LDMAR	unknown	Regulating normal pollen development	[205]
PMS1T	miR2118	Regulating photoperiod-sensitive male sterility.	[206]
Ef-cd	SDG724, OsSOC1	Increase the H3K36me3 level of the OsSOC1 and reducing the time-span for plant maturity.	[207]
MISSEN	HeFP	Regulating cytoskeleton polymerization during endosperm development.	[208]
CIRCRNA (SEP3)	SEP3	Driving floral homeotic phenotypes through exon-skipped AS variant of SEP3.	[209]
HID1	PIF3	Inhibiting hypocotyl elongation by modulating the chromatin structure of the promoter of PIF3.	[117]
AG-incRNA4	CLF, AG	Regulating leaf development through H3K27me3-mediated repression of AG.	[210]
TWISTED LEAF	OsMYB60	Regulating leaf blade flattening through chromatin modifications of OsMYB60.	[211]
APOLO	PID, WAG2, WRKY42	Regulating lateral root development through polar auxin transport by controlling chromatin loop dynamics and leading to low temperature-induced root hair elongation.	[129,212,213,214]
LAIR	LRK, OsMOF and OsWDR5	Regulating rice grain yield by chromatin modification and activating promoters of the LRKs gene.	[215]

## Data Availability

Not applicable.

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
