# Peer review of "The Characters of Non-Coding RNAs and Their Biological Roles in Plant Development and Abiotic Stress Response"

_ijms, 2022, doi:10.3390/ijms23084124_

Round 1
Reviewer 1 Report
Ma et al. have written a long and complex review on non-coding RNA and discussed their roles in plant development and abiotic stress response. It would be a great resource for educators if the manuscript were appropriately modified. Though the review is long, it is plain. In my modest opinion, an excellent review, on the one hand, should address the most advanced research outcomes in the field; On the other hand, it should address how those advances change our understanding of the field and how the review is different from previous/other ones. Unfortunately, the latter one is missed in the current version. For instance, a review article published in IJMS, titled "Plant Non-Coding RNAs: Origin, Biogenesis, Mode of Action and Their Roles in Abiotic Stress" has summarized the role of ncRNA in abiotic stress two years ago. Do the authors have different opinions regarding the role of ncRNA within the previous one? Have recent research outcomes in the past two years answered some of the critical/challenging questions raised before? Moreover, in the conclusion and perspectives, the author should also discuss the topics under the spotlight and the most challenging questions on the role of ncRNA in regulating plant development and abiotic stresses.
The main text of the review focused on miRNA and LncRNA. However, the authors have mentioned other ncRNAs, such as siRNA. Nevertheless, their roles were ignored. With that, the title of the review should be modified and narrowed. Furthermore, the second section title, "The classification and function of plant ncRNAs," should also be modified. "Function" is too broad here. The authors mainly discussed the classification, biogenesis, and action of the mode of ncRNAs.
The third section shall combine with the second section. I did not see reasons to separate the part from the previous one, especially for the LncRNA parts. The authors summarize the acting mode of LncRNAs, which are presented in Figure 2.
Plant development consists of many aspects in addition to the topics discussed. Therefore, I suggest the authors summarize the function of ncRNAs in shaping the fruit/grain size and maturation.
Since phytohormone pathways influence the expression of ncRNA, the model in Figures 3 and 5 might need some modification to show those effects. For example, in Figure 5, the miR847/IAA 28 pathway should point to the Auxin signaling, since the expression of miR847 was induced by auxin, referring to the manuscript "Cleavage of INDOLE-3-ACETIC ACID INDUCIBLE28 mRNA by microRNA847 upregulates auxin signaling to modulate cell proliferation and lateral organ growth in Arabidopsis."
The model in figure 6 illustrating pathways affecting the cold/heat stress needs to improve. I noticed that many miRNAs affect the HSP proteins of the host, which could be categorized together. This adjustment could help the reader understand the role of miRNAs clearer.
I suggest the author go through the whole manuscript and check the grammar mistakes, errors, and typos. Below, I just listed a few examples.
Line 46, "Brachypodium" should be italic.
Line102, the whole protein/domain name should be given. (The whole name of many genes are missing across the whole manuscript)
The full name of ceRNA should be given in Line 214, but not in line 216
Line 114, "MicroRNAs" should be abbreviated as "miRNAs".
Line 165, "almost 90% of the genome is transcribed by using high-throughput sequencing technology," should be "almost 90% of the genome was found be transcribed by using…"
Author Response
Reviewer1
Ma et al. have written a long and complex review on non-coding RNA and discussed their roles in plant development and abiotic stress response. It would be a great resource for educators if the manuscript were appropriately modified. Though the review is long, it is plain. In my modest opinion, an excellent review, on the one hand, should address the most advanced research outcomes in the field; On the other hand, it should address how those advances change our understanding of the field and how the review is different from previous/other ones. Unfortunately, the latter one is missed in the current version. For instance, a review article published in IJMS, titled "Plant Non-Coding RNAs: Origin, Biogenesis, Mode of Action and Their Roles in Abiotic Stress" has summarized the role of ncRNA in abiotic stress two years ago. Do the authors have different opinions regarding the role of ncRNA within the previous one? Have recent research outcomes in the past two years answered some of the critical/challenging questions raised before? Moreover, in the conclusion and perspectives, the author should also discuss the topics under the spotlight and the most challenging questions on the role of ncRNA in regulating plant development and abiotic stresses.
Response:
Thank you for your professional review and valuable suggestions to improve the quality of our manuscript.
We have revised the abstract and introduction, clarifying the recent progress and spotlight of the review focusing on. We have cited the review article "Plant Non-Coding RNAs: Origin, Biogenesis, Mode of Action and Their Roles in Abiotic Stress" and highlighted our key point in plant development and abiotic stresses. Please refer to the revised manuscript in p1 L17-23 and p2 L69-85. In conclusion and perspectives, we have added the discussion of further research “The further research field is how the development and stress signals are sensed to promote the expression of ncRNAs and transduced to specially regulate the transcrip-tion of target genes in different metabolic pathways”.
The main text of the review focused on miRNA and LncRNA. However, the authors have mentioned other ncRNAs, such as siRNA. Nevertheless, their roles were ignored. With that, the title of the review should be modified and narrowed.
Response:
Thanks for your advices. We have added the part of siRNA in the revised manuscript in p5 L217-236.
Furthermore, the second section title, "The classification and function of plant ncRNAs," should also be modified. "Function" is too broad here. The authors mainly discussed the classification, biogenesis, and action of the mode of ncRNAs. The third section shall combine with the second section. I did not see reasons to separate the part from the previous one, especially for the LncRNA parts. The authors summarize the acting mode of LncRNAs, which are presented in Figure 2.
Response:
Thank you for your comments. We have revised the title of the second part to “Classification and function action of plant ncRNAs” and integrated the third section into the second part.
Plant development consists of many aspects in addition to the topics discussed. Therefore, I suggest the authors summarize the function of ncRNAs in shaping the fruit/grain size and maturation.
Response:
Thank you for your valuable suggestions. We have summarized the the function of ncRNAs in shaping the fruit/grain size and maturation in p14 3.1.6.
Since phytohormone pathways influence the expression of ncRNA, the model in Figures 3 and 5 might need some modification to show those effects. For example, in Figure 5, the miR847/IAA 28 pathway should point to the Auxin signaling, since the expression of miR847 was induced by auxin, referring to the manuscript "Cleavage of INDOLE-3-ACETIC ACID INDUCIBLE28 mRNA by microRNA847 upregulates auxin signaling to modulate cell proliferation and lateral organ growth in Arabidopsis." The model in figure 6 illustrating pathways affecting the cold/heat stress needs to improve. I noticed that many miRNAs affect the HSP proteins of the host, which could be categorized together. This adjustment could help the reader understand the role of miRNAs clearer.
Response:
Thank you for your comments. We have revised the figure 3, figure 5 and figure 6 according to your suggestions.
I suggest the author go through the whole manuscript and check the grammar mistakes, errors, and typos. Below, I just listed a few examples.
Line 46, "Brachypodium" should be italic.
Line102, the whole protein/domain name should be given. (The whole name of many genes are missing across the whole manuscript)
The full name of ceRNA should be given in Line 214, but not in line 216
Line 114, "MicroRNAs" should be abbreviated as "miRNAs".
Line 165, "almost 90% of the genome is transcribed by using high-throughput sequencing technology," should be "almost 90% of the genome was found be transcribed by using…"
Response:
Thank you for your comments. We have revised the grammar mistakes, errors, and typos in the whole manuscripts. Thank you again.

Reviewer 2 Report
This is an interesting, although extremely complicated paper renewing various aspects of Non-coding RNA and their biological roles in plant development and abiotic stress. Part of the complication is that it is not clear, from the Introduction, as to what specific topics are covered in this paper. The authors have used past tenses - such as this subject has been described that topic has been reviewed - which makes it hard to figure out what is reported previously and what is being discussed in this paper
I find the last section (Conclusions) to be most informative and well written. If Introduction will tell the reader what to expect to read in this paper, it will make this complex (but important) paper easier to follow.
There are a number of incorrect English (mainly the use of wrong tenses, as mentioned above) and some awkward phrases that need to be corrected.
Author Response
Reviewer2
This is an interesting, although extremely complicated paper renewing various aspects of Non-coding RNA and their biological roles in plant development and abiotic stress. Part of the complication is that it is not clear, from the Introduction, as to what specific topics are covered in this paper. The authors have used past tenses - such as this subject has been described that topic has been reviewed - which makes it hard to figure out what is reported previously and what is being discussed in this paper
Response:
Thank you for your professional review and valuable comments. We have revised the abstract and introduction, clarifying the recent progress and spotlight of the review focusing on. Please refer to the revised manuscript in p1 L17-23 and p2 L69-85.
I find the last section (Conclusions) to be most informative and well written. If Introduction will tell the reader what to expect to read in this paper, it will make this complex (but important) paper easier to follow.
Response:
Thank you for your comments. We have revised the introduction in p2 L69-85.
There are a number of incorrect English (mainly the use of wrong tenses, as mentioned above) and some awkward phrases that need to be corrected.
Response:
Thank you for your comments. We have corrected the tenses in the whole manuscript. Thank you again.

Round 2
Reviewer 1 Report
My questions were addressed adequately in the revised version. The manuscript is logically organized well. I only find several grammar mistakes. For instance, the title of paragraphs 4.3 & 4.4 should be a sentence, as 4.1 and 4.2. "...are involved in ......". In addition, I suggest the authors polish the abstract. I give only a few samples here. In line 13, "the regulation of" can be replaced by "regulating". In Line 20, "with a focus on" can be substituted with "focusing on."